# The Big Five personality traits and the fear of COVID-19 in predicting depression and anxiety among Japanese nurses caring for COVID-19 patients: A cross-sectional study in Wakayama prefecture

Ryo Odachi[1]*, Shun Takahashi[2,3,4,5], Daichi Sugawara[6], Michiyo Tabata[3‡], Tomomi Kajiwara[1‡], Masaya Hironishi[7‡], Momoko Buyo[1]

1 Division of Health Sciences, Graduate School of Medicine, Osaka University, Suita City, Osaka, Japan,
2 Clinical Research and Education Center, Asakayama General Hospital, Sakai City, Osaka, Japan,
3 Department of Neuropsychiatry, Wakayama Medical University, Wakayama City, Wakayama, Japan,
4 Graduate School of Rehabilitation Science, Osaka Metropolitan University, Habikino City, Osaka, Japan,
5 Wakyamma Medical University Kihoku Hospital, Ito Gun, Wakayama, Japan, 6 Department of Psychiatry, Graduate School of Medicine, Osaka University, Suita City, Osaka, Japan, 7 Faculty of Human Sciences, University of Tsukuba, Tsukuba City, Ibaraki, Japan

☯ These authors contributed equally to this work.
‡ MT, TK and MH also contributed equally to this work.
* odachi@sahs.med.osaka-u.ac.jp

## Abstract

Recent studies have found a relationship between fear of COVID-19 and mental health problems. Medical workers caring for COVID-19 patients tend to suffer from mental health problems; however, the impact of their personality traits, in the form of mental problems like depression and anxiety in Japan is unclear. In this study, we investigated the risk of nurses' depression and anxiety, predicted by the fear of COVID-19 and the Big Five personality traits. A total of 417 nurses working in hospitals providing care to COVID-19 patients in Wakayama prefecture of the Kansai region participated in this study. The questionnaires comprised items on nurses' basic characteristics and three scales: the Fear of COVID-19 Scale 2020, the Big-Five Scale, and the Japanese version of the Hospital Anxiety and Depression Scale (HADS). Depression and anxiety in the HADS were set as dependent variables, and basic attributes, fear, and personality traits as independent variables; multivariate logistic regression analyses were conducted. The questionnaire, with no missing items was distributed from February to March 2021. Neuroticism (OR = 1.06, 95%CI = 1.03–1.09) was the only significant factor associated with the depression symptom, and both FCV-19S scores (OR = 1.16, 95%CI = 1.09–1.23) and neuroticism (OR = 1.09, 95%CI = 1.06–1.13) were the significant factors associated with anxiety. The Nagelkerke's R squared was 0.171 in the depression model and 0.366 in the anxiety model. Thus, it was found that it is necessary to support nurses' mental health by developing methods suitable to their personalities.

**Data Availability Statement:** All relevant data are within the paper and its Supporting information files.

**Funding:** This study was funded by the 2020 Wakayama Medical University Special Grant-in-Aid for Research Projects (20TS04) to ST. The funders had no role in study design, data collection and analysis, decision to publish, or preparation of the manuscript.

**Competing interests:** The authors have declared that no competing interests exist.

# Introduction

Nurses play an important role in maintaining healthcare systems especially during difficult times like the COVID-19 pandemic caused by the SARS-CoV-2 virus [1]. A study found that nurses on the front-lines had a high level of fear of COVID-19 [2]. Furthermore, nurses are required to use stricter infection prevention than usual, which could increase their workload and the tension associated with it. Even in their daily lives, nurses have been required to limit their activities strongly to stop the spread of the virus, and thus, they must have faced a significant amount of tension for more than one year [3]. Since about April 2021, a year after the pandemic outbreak in Japan, the media frequently reported that the COVID-19 surge was putting a strain on hospitals, which in addition to the tension in society, also led to an increase in the mental and physical burden on nurses [4–6]. Therefore, the risk of burnout among nurses may increase with the protracted pandemic.

In Greece, it was found that a high level of fear about COVID-19 in the general population predicts generalized anxiety disorder and post-traumatic stress disorder [7]. In China during the pandemic, it was reported that 50% of healthcare workers had depressive symptoms, 45% had anxiety symptoms, 34% had insomnia symptoms, and 72% had distress symptoms [8]. It was also reported that frontline nurses' fear of COVID-19 predicted their mental health problems in Pakistan [9]. A meta-analysis examining the psychological responses associated with outbreaks of infectious diseases such as SARS, MARS, pandemic influenza, Ebola hemorrhagic fever, and COVID-19 also found increased trauma and psychological distress among healthcare workers [10]. Thus, in various countries, a relationship between fear of COVID-19 and mental health problems among nurses and the general public was observed.

In Japan, there were five waves of COVID-19 and four declarations of a state of emergency by the end of 2021 (Fig 1). The maximum number of new patients afflicted with COVID-19 per day was about 25,000, and 20,000 people have died from the virus so far in the country [11]. During the fourth and fifth waves of the pandemic, there were many reports of hospital beds being scarce, which meant that people could not be hospitalized even if they had symptoms of COVID-19, leading to a heightened sense of social tension [12, 13]. Previous studies have shown that more than 20% of healthcare workers in a tertiary hospital met the burnout criteria in the first wave of the pandemic [14]. Healthcare workers suffer from greater fear and psychological distress from COVID-19 than the general public [15]. In a study, it was reported that between 20% and 30% of nurses tending to the COVID-19 patients were in a state of high mental distress, and the increase in the number of patients was a factor affecting their mental health problems and the intention to resign [6]. Another study in a university hospital showed that approximately 30% of young nurses with depression showed depressive symptoms during the pandemic [16]. Nurses in closer contact with patients are considered more likely to have mental health problems than other healthcare employees. Additionally, Japan has a cultural uniqueness wherein there is a strong sense of belongingness among people to their community or group and they are unlikely to accept any behavior that deviates from their own norms [17]. While this can be beneficial as people can voluntarily take precautions such as wearing masks without legal restrictions, it can also increase the risk of being infected. Ohue et al. reported that damage from harmful rumors increases the severity of mental health problems [6]. Therefore, social pressure might have a direct influence on Japanese nurses' fear of COVID-19 and mental health problems.

Personality traits are predictors of important health outcomes as they impact physical diseases and affect mental health of an individual. For instance, studies found that there were relationships between personality and mental health problems, such as depression, anxiety disorder, addiction, low level of happiness, alexithymia, and burnout [18–23]. In their

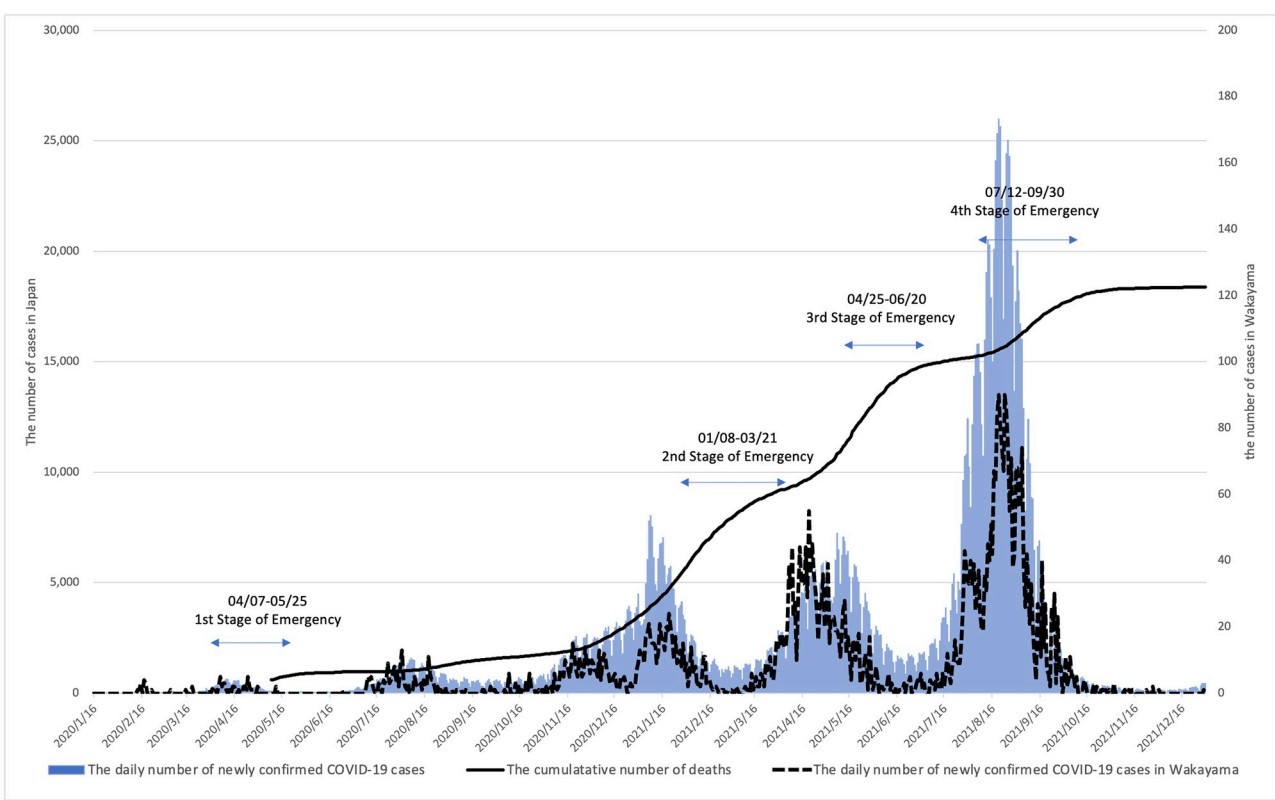

**Fig 1. The changes in reporting related to COVID-19 and the timing of the state of emergency in Japan and Wakayama prefecture.** Data source: https://www.mhlw. go.jp/stf/covid-19/kokunainohasseijoukyou.html.

diathesis-stress model, Metalsky et al. showed that even when the stressor is the same among a wide range of people, the response varies from depending on the person who perceives it [24]. This implies that the current pandemic is considered to be a major obstacle and stress for nurses; however, its impact varies from person to person. Therefore, to explore the details of the pandemic's impact on mental health, it is necessary to focus on the individual's personality, which is considered to be one of the major factors of individual differences. In the Japanese COVID-19 pandemic scenario, no study has addressed the relationship between nurses' mental health and personality traits. It is also thought that understanding the characteristics and impacts of nurses' personalities will enable more personalized mental health support for each nurse. This study adopted the Big Five personality traits, which is one of the most popular personality models used in many studies [25–27]. It consists of five higher-order personality traits: extraversion, neuroticism, openness, conscientiousness, and agreeableness, and it captures an individual's personality comprehensively and simply.

The purpose of this study was to investigate the risk of depression and anxiety in nurses who had to deal with patients with COVID-19 or were suspected to have COVID-19, predicted by the fear of COVID-19 and the Big Five personality traits. A multivariate logistic regression analysis was performed to assess nurses' risk for depression and anxiety. This method was considered not to increase depression or anxiety scores but to identify factors that influence the presence or absence of the symptoms.

## Materials and methods

### Participants and procedures

In this cross-sectional study, nurses experienced in caring for COVID-19 patients or suspected COVID-19 patients were included. The inclusion criteria were (1) nurses from 13 hospitals that had separate beds for infectious disease for treating COVID-19, in June 2020 in Wakayama prefecture and who agreed to cooperate in the investigation with permission from the hospital directors; and (2) nurses who had an experience in caring for COVID-19 patients, including possible cases; and (3) agreed to cooperate with the mode of conducting the study (anonymous self-administered questionnaires). No exclusion criteria were defined. Before distributing the questionnaires, the researchers obtained information from each hospital about the number of nurses who had experience with COVID-19-related work and sent the required number of questionnaires to each hospital. The principal researcher requested the cooperation of nursing directors at each hospital, in writing and verbally; the questionnaires were distributed to individual participants by nursing managers, which were then dropped into the collecting box at each hospital by participants.

Data were collected from February 22 to March 12, 2021 in the Wakayama prefecture of Japan, which is adjacent to one of the principal Japanese cities, Osaka. The population of Wakayama is approximately one million and their aging rate is 32%, which is relatively high. When this study was conducted, the number of new positive cases of COVID-19 was less than 10 per day, and the utilization of hospital beds, which peaked in mid-January, also decreased [28]. This period is the time when the third wave of infection was contained, and the state of emergency was lifted by the end of February. However, vaccination for medical workers and the older people had not yet begun.

The participants provided written informed consent before their participation; they could freely decline participation in the survey at any time. They were neither compensated nor rewarded for participation. The privacy of the participants was protected. The survey was anonymous, and only the researcher had access to the data. The participants agreed to the study by submitting the completed questionnaire. This study was approved by The Research Ethics Committee of Wakayama Medical University (approval no. 3006).

### Measures

The questionnaire included questions on basic characteristics, including sex, age, years of experience as a nurse, presence of family members living together, and willingness to provide care for COVID-19 patients. It comprised three additional scales: the Japanese version of the Fear of COVID-19 Scale (FCV-19S) [29], the Big Five Scales (BFS) [30], and the Japanese version of the Hospital Anxiety and Depression Scale (HADS) [31].

The Japanese version of the FCV-19S was used to assess nurses' fear of COVID-19. This scale consists of seven items rated on a 5-point Likert scale (1 = "*strongly disagree*"; 5 = "*strongly agree*"). Possible scores range from 7–35, with a higher score indicating a stronger fear of COVID-19. The total scores were used in the analysis. This FCV-19S was previously validated among the general population and the internal consistency value of FCV-19S was .83 [29]. The FCV-19S was also used in the previous studies for nurses [2, 32].

The BFS, developed by translating the Adjective Checklist (ACL) [33] and extracting its items, was used to investigate personality traits. It comprised 60 items to be rated on a 7-point Likert-type scale (1 = "*does not describe me at all*" to 7 = "*totally describes me*") on each of the five personality trait factors: extraversion (outgoing/energetic vs. solitary/reserved), neuroticism (sensitive/nervous vs. secure/confident), openness (inventive/curious vs. consistent/cautious), conscientiousness (efficient/organized vs. easy-going/careless), and agreeableness

(friendly/compassionate vs. challenging/detached). Each total score (the possible score range is 7–84) was used in the analysis. The BFS was previously reported to have acceptable internal consistency (a = .91, .92, .86, .88, and .84 for extraversion, neuroticism, openness, conscientiousness, and agreeableness, respectively) and is frequently used in studies conducted in Japan [30, 34]. The BFS was previously validated [35] and used in studies for a wide range of ages.

The HADS is a scale that evaluates the status of depression and anxiety [36]. The Japanese version of HADS used in this study was established for reliability and validity (a = .77, .79; depression, anxiety) [37]. Furthermore, prior studies validated this scale among the general population and patients [38], and nurses [39–41]. The HADS comprises seven items related to anxiety and seven related to depression rated on a 4-point scale (0–3), and high scores represent more frequent or stronger symptoms. For example, in the question "Worrying thoughts go through my mind", 0 represents "Only occasionally" and 3 represents "A great deal of the time". Contrarily, in the question "I can sit at ease and feel relaxed", 0 means "Definitely" and 3 means "Not at all". The possible scores range, which is the total of each of the seven items, is 0–21. A total score of over 8 points serves as a cut-off point for both the depression subscale and anxiety [31], which indicates the presence of symptoms of depression or anxiety.

## Statistical analyses

All data analyses were performed using the open-source statistical framework of R [42] on the open-source user interface RStudio [43]. Descriptive statistics were analyzed for all basic characteristic data and study variables. The mean and standard error (SE) were calculated for each continuous variable. The distribution of scores around the mean was analyzed using skewness and kurtosis tests. Both kurtosis and skewness met the standard value ($< \pm 2.00$) [44] and the normality of each item was confirmed. The frequency was calculated for each categorical variable.

The Pearson Product-Moment correlation was calculated to assess the model by revealing the relationships between all the continuous variables in this study. For each model, the Variance Inflation Factor (VIF) of all the variables were confirmed for multicollinearity. A prior study speculated that correlation coefficients of 0.8 to 0.9 between the independent variables could cause problems [45] and that there was a significantly strong correlation between age and years of experience as a nurse. Therefore, variables related to experience were excluded from the models.

Finally, multivariate logistic regression analysis (forced entry method) was used to evaluate the risk of the independent variables in the presence of symptoms of depression (Model A) and anxiety (Model B). The cut-off point of HADS was used to determine the presence of depression and anxiety symptoms (total scores of each subscale >8). Thereafter, the relationship between anxiety and depression as dependent variables, age and sex as control variables, and FCV-19S and Big Five factors as independent variables, were examined. Age and sex are often reported to affect the Big Five personality traits [46, 47]; for example, honesty increases with age [48]; thus, they were selected as control variable in this as well as in prior studies. The odds ratios (OR) and 95% confidence intervals (CI) were calculated. The goodness of fit was estimated using Hosmer-Lemeshow test. Nagelkerke's R squared which indicates the power of explanation of the model was calculated. The level of statistical significance was set at $p < 0.05$.

## Results

### Demographics

A total of 525 participants responded to the questionnaires, and data was obtained from 417 participants who responded with no missing values (effective response rate = 66.30%). Table 1 presents the demographic characteristics of the participants.

**Table 1. Participant demographics (n = 417).**

| Variables | mean±SE |
|---|---|
| Age (years) | 40.77±0.57 |
| Experience as a Nurse (years) | 15.63±0.53 |
| HADS scores for Depression (total score) | 4.56±0.26 |
| HADS scores for Anxiety (total score) | 4.96±0.17 |
| FCV-19S (total score) | 20.02±0.18 |
| Extraversion (total score) | 52.35±0.04 |
| Neuroticism (total score) | 52.04±0.05 |
| Openness (total score) | 46.01±0.04 |
| Conscientiousness (total score) | 49.32±0.04 |
| Agreeableness (total score) | 53.90±0.04 |
| Variables | n (%) |
| Sex | |
| Male | 38 (9.11%) |
| Female | 379 (90.89%) |
| COVID-19 diagnosis of the patient under their care | |
| Confirmed | 320 (76.74%) |
| Suspected | 97 (23.26%) |
| Willingness to provide care | |
| Yes | 37 (8.87%) |
| No | 380 (91.13%) |
| Living with family | |
| Yes | 333 (79.86%) |
| No | 84 (20.14%) |
| Presence of depression | |
| No (< 8) | 337 (80.82%) |
| Yes (≧ 8) | 80 (19.18%) |
| Presence of anxiety | |
| No (< 8) | 302 (72.42%) |
| Yes (≧ 8) | 115 (27.58%) |

SE: standard error

FCV-19S: Fear of COVID-19 Scale.

HADS: Hospital Anxiety and Depression Scale

COVID-19: Coronavirus Disease 2019

The mean value of FCV-19S was 20.02.81 (SE = 0.18; range = 7–35), that of depression and anxiety were 4.56 (SE = 0.26; range = 0–16) and 4.96 (SE = 0.17; range = 0–14), respectively. The number of people with depression and anxiety was 80 (19.18%) and 115 (27.58%), respectively. The relationships between all the variables in this study are presented in Table 2.

Table 3 presents the multivariate logistic regression analysis results for models A and B. Model A revealed that only neuroticism (OR = 1.06, 95%CI = 1.03–1.09) was the significant factor associated with depression. Model B revealed that age (OR = 1.03, 95%CI = 1.00–1.05), sex (OR = 2.93, 95%CI = 1.09–9.08), FCV-19S scores (OR = 1.16, 95%CI = 1.09–1.23) and neuroticism (OR = 1.09, 95%CI = 1.06–1.13) were the significant factors associated with anxiety. The results of the Hosmer-Lemeshow test showed that the models adequately fit the data (Model A; $X^2$ = 10.395, df = 8, p = 0.2384, Model B; $X^2$ = 13.654, df = 8, p = 0.0912). Model A explained 17.1% (Nagelkerke R-squared) of the variance for depression and model B explained 36.6% (Nagelkerke R-squared) of the variance for anxiety.

**Table 2. Correlation matrix between each variable (N = 417).**

| | 1. Age | 2 | 3 | 4 | 5 | 6 | 7 | 8 | 9 |
|---|---|---|---|---|---|---|---|---|---|
| 2. Experience as a nurse | 0.85*** | | | | | | | | |
| 3. FCV-19S | 0.02 | 0.03 | | | | | | | |
| 4. Depression | 0.08 | 0.05 | 0.30*** | | | | | | |
| 5. Anxiety | 0.13** | 0.10* | 0.45*** | 0.61*** | | | | | |
| 6. Extraversion | 0.05 | 0.07 | -0.15** | -0.27*** | -0.20*** | | | | |
| 7. Neuroticism | -0.03 | -0.04 | 0.34*** | 0.43*** | 0.54*** | -0.46*** | | | |
| 8. Openness | 0.01 | 0.03 | 0.02 | -0.13** | -0.06 | 0.46*** | -0.19*** | | |
| 9. Conscientiousness | 0.07 | 0.04 | -0.04 | -0.21*** | -0.21*** | 0.23*** | -0.23*** | 0.24*** | |
| 10. Agreeableness | -0.02 | -0.03 | -0.13** | -0.22*** | -0.26*** | 0.34*** | -0.37*** | 0.19*** | 0.45*** |

FCV-19S = Fear of COVID-19 Scale.

* $p < 0.05$.

** $p < 0.01$.

*** $p < 0.001$.

# Discussion

This study aimed to assess the risk of depression and anxiety among nurses who have experience of caring for patients with COVID-19 in Japan. The results of this study indicated that

**Table 3. Multivariate logistic regression analysis results (N = 417).**

| Variables | OR | SE | 95%CI | | | p-value | VIF |
|---|---|---|---|---|---|---|---|
| Model A: Model A: Age + Sex + FCV-19S + Big Five Personality traits = > Depression | | | | | | | |
| Age | 1.01 | 0.01 | 0.98 | to | 1.03 | 0.5 | 1.02 |
| Sex (Male = 0, Female = 1) | 2.37 | 0.57 | 0.86 | to | 8.41 | 0.13 | 1.01 |
| FCV-19S | 1.06 | 0.03 | 1.00 | to | 1.12 | 0.056 | 1.10 |
| Extraversion | 0.99 | 0.02 | 0.96 | to | 1.02 | 0.4 | 1.48 |
| Neuroticism | **1.06** | 0.02 | 1.03 | to | 1.09 | **<0.001** | 1.29 |
| Openness | 1.01 | 0.02 | 0.98 | to | 1.05 | 0.6 | 1.32 |
| Conscientiousness | 0.97 | 0.02 | 0.94 | to | 1.01 | 0.1 | 1.23 |
| Agreeableness | 1.00 | 0.02 | 0.97 | to | 1.04 | >0.9 | 1.28 |
| Negelekerke R-sq = 0.171 | | | | | | | |
| Hosmer and Lemeshow goodness of fit test: x-squared (8) = 10.395, p = 0.2384 | | | | | | | |
| Model B: Model A: Age + Sex + FCV-19S + Big Five Personality traits = > Anxiety | | | | | | | |
| Age | **1.03** | 0.01 | 1.00 | to | 1.05 | **0.021** | 1.03 |
| Sex (Male = 0, Female = 1) | **2.93** | 0.54 | 1.09 | to | 9.08 | **0.045** | 1.03 |
| FCV-19S | **1.16** | 0.03 | 1.09 | to | 1.23 | **<0.001** | 1.06 |
| Extraversion | 1.02 | 0.02 | 0.99 | to | 1.05 | 0.2 | 1.50 |
| Neuroticism | **1.09** | 0.02 | 1.06 | to | 1.13 | **<0.001** | 1.25 |
| Openness | 1.01 | 0.02 | 0.98 | to | 1.05 | 0.5 | 1.29 |
| Conscientiousness | 0.98 | 0.02 | 0.95 | to | 1.01 | 0.2 | 1.26 |
| Agreeableness | 0.98 | 0.02 | 0.94 | to | 1.01 | 0.2 | 1.31 |
| Negelkerke R-sq = 0.366 | | | | | | | |
| Hosmer and Lemeshow goodness of fit test: x-squared (8) = 13.654, p = 0.0912 | | | | | | | |

Fear of COVID-19 Scale. OR = odd ration. SE = standard error. 95%CI = 95% confidence interval.

nurses' risk of anxiety was related to their age, fear of COVID-19, and neuroticism, whereas their risk of depression was related to only neuroticism.

Although fear is usually a temporary emotion and a normal reaction to a real or imagined threat, it is distinguished as a pathological fear if it is long-term or significantly interferes with daily life [49]. Furthermore, it has been reported that this fear could cause anxiety and depression in majority of the population [50]. The COVID-19 pandemic increased mental health problems including depression and anxiety in general population [51–53]. Some studies reported a higher rate of depression and anxiety in healthcare workers than the general population [54–56], and the FCV-19S scores predicted risk for depression and anxiety [32, 57]. Koiwa et al. [2] reported that Japanese nurses scored higher on the FCV-19S compared to nurses in other countries, and while they said it was not possible to compare the scores between each study, their mean was higher than some proposed FCV-19S cutoff values [7, 58]. In the present study, the percentage of nurses with anxiety scores higher than the HADS cutoff was about 30% of the total number of nurses; further, the percentage of those with depression scores exceeding the cutoff was 20% of the total number of nurses. Although the frequency of people with depression and anxiety varied from study to study, several studies reported that anxiety is more strongly associated with FCV-19S scores than depression [59–61]. These results were consistent with this study. The results of this current study suggest that nurses in Japan may be at higher risk for mental health problems and that it is necessary to construct more effective support systems for them.

Additionally, it has been reported that direct contact with COVID-19 patients (including suspected ones) was a risk factor for anxiety but not for depression among nurses [62], and the participants in this study were all nurses with experience in care for COVID-19 patients, which may have been a factor of the low frequency of depression. Contrarily, Xia et al. [63] reported that nurses and physicians on the front-line had lower levels of anxiety than other hospital staff. However, some studies showed that being on the front-line may cause anxiety and mental burden [8, 54]. Therefore, further study is required to explore the impact of the different levels of how the nurses have experienced and contributed to the care of COVID-19 patients. The fear of COVID-19 can be linked to the state of society, and it is possible that a prolonged coexistence with this virus may lead to chronic fear among the nurse population, which can worsen their depression and anxiety. Furthermore, since April 2021, when data collection for this study was completed, the outbreak of COVID-19 in western Japan, including Wakayama, had become more severe with inadequate vaccination. As a result, nurses may have been under higher levels of stress. A longitudinal study would be required to clarify how this serious situation affects nurses' mental health.

In this study, we found associations between nurses' mental health problems and neuroticism in the Big Five personality traits. Many prior studies have associated neuroticism with depression and anxiety [18, 64–67], and recent studies during COVID-19 also presented the relationships between neuroticism and mental health problems [68–71]. Neuroticism represents the tendency of individuals to experience negative emotions, including depression, anxiety, and anger [72]. Individuals with high neuroticism scores have some tendencies, such as difficulty in coping with stressful experiences and susceptibility to negative emotions in general [73]. Therefore, it is considered that high neuroticism is associated with various mental health problems, including depression and anxiety [65, 66, 74], and similar results were also presented in some studies that examined healthcare workers during the COVID-19 pandemic [75, 76]. The results of the present study are in line with these prior studies that found that nurses' anxiety and depression are associated with their high neuroticism.

Although many studies reported that extraversion was also associated with anxiety and depression [64, 65, 77–79], the results in the present study were not consistent with them.

Individuals with high extraversion scores have certain tendencies, such as great activity, need for stimulation, seeking social interaction, and experiencing positive emotions [80]. According to a previous study, they are more capable of adjusting to life-changing events to use adaptive strategies [81], and a low extraversion score was reported as a risk factor for depression and anxiety [64, 69, 82]. Different from neuroticism, a variety of views exist regarding the impacts on depression and anxiety related to extraversion. For example, research by Jorm et al. [67] expressed doubt about the generalizability of a synergistic effect between high neuroticism and low extraversion in predicting depression and anxiety, which other studies had presented. Some other studies presented that extraversion was only weakly related to the risk for depression and anxiety [77, 83] and mediated them while, neuroticism was directly linked to the risk for depression and anxiety [70], with differences in the impact based on age [78]. Although high extraversion has positive impacts on mental health as described above, a study reported that it might increase stress due to the inability to socialize during the pandemic [68]. Although there were no significant impacts of extraversion in the present study, this may be due to the impacts of the pandemic. The pandemic has caused movement restrictions, thereby making it necessary for people to avoid contact with each other. This may have brought about inadequate coping strategies for people with high extraversion. Therefore, extraversion is considered to not have positively influenced nurses' risk for depression and anxiety. There may be various mediators in the relationship between each factor of Big Five personality traits and repression/anxiety. However, further research is required to explore these impacts. Additionally, it is necessary to establish a system for early screening and treatment because an elevated level of neurotic tendencies can increase the possibility of inappropriate coping with excessive stress. In recent years, some studies have focused on the development of programs for general population using games based on personality traits and effective ways of communicating information according to them [84, 85]. Based on these findings, it is necessary to expand educational resources to maintain and improve the mental health of nurses.

Older age was a risk factor for anxiety in the present study. One study reported lower anxiety and depression in nurses with more experience [86]; however, these results differ from that of the present study. In China, it was reported that those with more than 10 years of clinical experience faced increased risk of anxiety and depression [87] and as per this study, the reason for this might be due to traditional familial responsibilities, such as caring for family members living together, along with occupational fatigue. In Japan, the influence of traditional family roles remains strong, which may have resulted in similar findings. Additionally, in various countries, it has been reported that females are more likely to experience mental health problems such as fear, depression, anxiety, and burnout during the COVID-19 pandemic [8, 56, 88–90], which is reflective of the results of the present study. However, the proportion of male participants in this study was less than 10%, and the 95% confidence intervals for the odds ratios were large, leaving the possibility of overestimating the effect of sex.

## Study limitations

The present study had several limitations. First, the BFS was used to investigate nurses' personality traits. The Big Five model is useful for investigating individual personality comprehensively and easily. However, this model has no specific background theory and physiological basis. Therefore, the use of the Big Five model for clinical diagnosis has been criticized partly as the factors in the model are clinically heterogeneous [91, 92]. For example, the Temperament and Character Inventory-Revised [91] and Zuckerman-Kuhlman Personality Questionnaire [93] are scales that can also measure individual personality traits, and these scales were developed from a psychobiological foundation. The findings were consistent with results of

molecular and genetic research [94–96]. Additionally, Cloninger's model can explain individual personality from two domains: temperaments that are relatively stable across the life span [97] and characters that are changeable to affect the environment and interaction with others through life [98]. Therefore, the model helps to explain the relationships between individual personality and physiological phenomena and psychiatric disorders from these two domains or specific profiles that consist of these factors robustly [99, 100]. Although the use of the scales in this study might have enabled clearer explanations, these scales consist of more than 100 items, and researchers are worried that this would be a burden on participants who were just caring for COVID-19 patients. Therefore, the BFS, which can respond easily, was used in the present study. Second, although the questionnaires used in the present study were conducted anonymously, it is possible that nurses may answer with social desirability, which may have biased the results because the nursing administrator distributed the questionnaire. Third, as this study was conducted only in the Wakayama Prefecture, which was one of the least COVID-19 infected provinces, it might have affected the FCV-19S and the HADS scores. Kito et al. showed that in Japan, where relational mobility is low, a small number of people with COVID-19 might face an increased fear of COVID-19 [101]. Additionally, it was said that there were some differences in personality traits due to regional differences in Japan [102]. The results of this study may not always be consistent with those of nurses in large cities of Japan, and the possibility of sampling bias cannot be ruled out. Fourth, as the participants in the study were nurses who responded to our questionnaire without missing values, there may be biases of personality measured. The small number of male participants is also one of the limitations in this study. Additionally, the number of participants across hospitals was unbalanced, and the differences in nurses' experiences between hospitals were likely not reflected in these results. Finally, a longitudinal investigation is required to explore the impact of pandemics on nurses' mental health since the present study utilized cross-sectional data, and the vaccination which started after our investigation can be affected nurses' mental health.

## Conclusions

The study found that neuroticism in the Big Five personality traits was a common risk factor for depression and anxiety determined by the HADS. The fear of COVID-19 was a risk factor only for anxiety in nurses caring for COVID-19 patients; however, the logistic regression analyses predicted that depression and anxiety explained about 20% and 40% of the variance, respectively. In previous studies, neuroticism and extraversion in the Big Five were associated with depression and anxiety. However, only neuroticism was related to these mental health problems in the present study. It was possible that the special situation of the pandemic affected these findings. Therefore, it is necessary for nurses caring for COVID-19 patients to support their mental health early and to develop methods that help them care for their mental health as per their personalities.

## Supporting information

**S1 Data.**
(XLSX)

## Acknowledgments

The authors are grateful to all the nurses who willingly participated in this study and Dr. Keii-chiro Adachi and Dr. Hironori Yada for their valuable advice.

## Author Contributions

**Conceptualization:** Ryo Odachi, Shun Takahashi, Daichi Sugawara, Michiyo Tabata, Momoko Buyo.

**Data curation:** Ryo Odachi, Shun Takahashi, Michiyo Tabata, Tomomi Kajiwara, Masaya Hironishi, Momoko Buyo.

**Formal analysis:** Ryo Odachi, Daichi Sugawara.

**Funding acquisition:** Shun Takahashi.

**Investigation:** Shun Takahashi, Masaya Hironishi, Momoko Buyo.

**Methodology:** Ryo Odachi, Shun Takahashi, Daichi Sugawara, Michiyo Tabata, Tomomi Kajiwara, Masaya Hironishi.

**Project administration:** Shun Takahashi, Masaya Hironishi, Momoko Buyo.

**Supervision:** Shun Takahashi, Daichi Sugawara, Masaya Hironishi.

**Validation:** Shun Takahashi, Daichi Sugawara, Michiyo Tabata, Momoko Buyo.

**Visualization:** Ryo Odachi, Shun Takahashi, Daichi Sugawara, Michiyo Tabata, Tomomi Kajiwara, Momoko Buyo.

**Writing – original draft:** Ryo Odachi.

**Writing – review & editing:** Shun Takahashi, Daichi Sugawara, Michiyo Tabata, Tomomi Kajiwara, Masaya Hironishi, Momoko Buyo.

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
