## [Decision Letter · Decision Letter 0]

1 Apr 2022

PONE-D-22-04225The Big Five personality traits and the fear of COVID-19 predict mental health problems in Japanese nurses caring for COVID-19 patients: A cross-sectional study in Wakayama prefecture.PLOS ONE

Dear Dr. Ryo Odachi, 

Thank you for submitting your manuscript to PLOS ONE. After careful consideration, we feel that it has merit but does not fully meet PLOS ONE’s publication criteria as it currently stands. Therefore, we invite you to submit a revised version of the manuscript that addresses the points raised during the review process.

 In addition to the response of the reviewers, authors must respond to the following comments. Plus, the paper needs thorough English editing by a professional editor or native English speaker focusing not only the grammar but also structure and organization of the writing.

We look forward to receiving your revised manuscript.

Kind regards,

Fakir Md Yunus, PhD, MSC, MPH, MBBS

Academic Editor

PLOS ONE

Journal Requirements:

Additional Editor Comments:

The Big Five personality traits and the fear of COVID-19 predict mental health problems in Japanese nurses caring for COVID-19 patients: A cross-sectional study in Wakayama prefecture.

PONE-D-22-04225

General comments:

This is an interesting paper, and the authors rightly pointed the one of key issues among the front-line health care professional during the COVID-19 pandemic. However, there are several major areas that authors can reconsider to increase the readability of the paper.

Abstract:

• Authors mentioned that “Our results showed an association between the HADS score and fear of COVID-19 and personality traits”- please consider mentioning whether this association is positive or negative.

• Also, authors may consider mentioning the beta estimates with CI of multiple regression model.

Introduction:

• Citations are missing in several statements in the introduction. Please provide citations on all the facts.

• Fig 1 is blur, therefore losses readability.

Methods:

• What is the study design?

• Please provide inclusion and exclusion criteria.

• This study was approved by the Research Ethics Committee of Z University (approval no. zzzz). Does this University exist? The ref. number does not look real.

• Please provide the Cronbach alpha of all measures used in this study.

Results:

• Please split the table 3 into two separate table since it has two different outcomes.

• Authors may not require repeating the results what was shown in the tables. I’d suggest just writing the key results presented in the table.

• I’m confused to understand the variables when I look at the table 1 and table 2. I see that authors presented categorical anxiety and depression scale in table 1 and continuous value in table 2. Please clarify.

Discussion:

• Authors need to elaborately discuss the study findings including clearly stating the interpretation of the results and what was found in the earlier studies.

• Furthermore, discussion is warranted based on theoretical aspect.

• I’d not use ‘mental health’ in the discussion rather use anxiety and depression.

• There are quite a number of limitations that authors should mention. For instance, generalizability, limitations of using self-reported data, causality...potential biases etc.

Conclusion:

• Authors mentioned that “The results of this study indicated that the fear of COVID-19 and the Big Five personality traits could predict the levels of depression and anxiety using the HADS.” – I’m not sure what this means?

o I understand the fear of covid-19 increases anxiety and depression among the Nurses? Is this correct? If yes, please provide a clear recommendation rather than using ambiguous terms.

o I’m lost about the conclusion on the Big Five personality traits. I am not sure what authors are trying to say? Is it that those 5 personality traits predict increased anxiety and depression, sorry if I got it wrong? Please help me understand the conclusion.

Reviewers' comments:

Reviewer's Responses to Questions

**Comments to the Author**

1. Is the manuscript technically sound, and do the data support the conclusions?

Reviewer #1: No

Reviewer #2: Yes

2. Has the statistical analysis been performed appropriately and rigorously? 

Reviewer #1: No

Reviewer #2: I Don't Know

3. Have the authors made all data underlying the findings in their manuscript fully available?

Reviewer #1: Yes

Reviewer #2: Yes

4. Is the manuscript presented in an intelligible fashion and written in standard English?

Reviewer #1: No

Reviewer #2: Yes

5. Review Comments to the Author

Reviewer #1: The manuscript contains a study that may be of interest to the Journal, but the authors must carry out an in-depth review of this version. Below are my main concerns and suggestions:

- The abstract contains excessive information on the method (questionnaires distributed, answers received, ...), only the final number of participants should be indicated since the rest of the information is already exposed in the manuscript. Instead, the implications and/or interest of the results obtained should be added.

- The use of the keyword “mental health”, even in the title, is too broad. With it, it is expected to find the evaluation of symptomatology corresponding to more disorders, but the manuscript is limited to exploring depression and anxiety. It is recommended to specify the symptomatology studied.

- The authors comment that in Wakayama prefecture the incidence of Covid-19 is low and detail the third wave situation at the time of collecting information (page 9, lines 110-121), this information can be reduced without losing the content essential.

- The use of Big Five personality, although widespread in the international community, is based on lexical aspects. There are other questionnaires with a robust psychobiological basis that are more suitable in the field of mental health work. Among them, Zuckerman's ZKPQ and Cloninger's TCI-R stand out. In the limitations section, this limitation should be added with the inclusion of the paper by Muro et al. (Chronobiology International, 28 (7), 690-696) in normal population and from Rio-Martinez et al. (Journal of Clinical Medicine, 9(6): 1876) in patients with mental disorders, the reading of both works in relation to the traits of personality of both questionnaires will provide the authors with this vision. The interest of considering more aspects of temperament or character, in the case of the TCI-R, is of the utmost importance. Introducing this approach also makes it possible to criticize the fact that the use of the Big Five personality presents relationships that are nonspecific in relation to the symptomatology studied.

- Measures. In this section, mention should not only be made of the reliability published in previous works, but also the detail of the reliability in the sample studied (FCV-19S, BFS and HADS) is required.

- In reference to the cut-off points of the HADS, the interest is focused on differentiating between “non-anxiety or depression” (0-7) and “possible case” (8 or more). Actually, the authors do not then work in their analyzes with the three possible groups indicated in the measures section (page 11). In table 1 it is not correct to define the depression and anxiety scores as low / high, I suggest adopting presence/absence of symptoms or yes/no possible case.

- The choice of linear regression analysis is not the most appropriate. To establish the existence or not of an association between personality traits and symptoms of anxiety and depression (presence/absence or yes/no possible case), the approach requires logistic regressions. Since concluding that, for example, the more neuroticism the score is higher, it does not mean that a high neuroticism is really associated with scores for the presence of symptoms.

- In the results text (pages 14-15) the numerical data already provided in Table 1 are repeated. The two paragraphs should be rewritten in a more qualitative way and only highlighting some data of more interest.

- Presentation of data in the tables can be improved. In Table 2 the first line can be deleted and the results could be better interpreted if the full name of the variables is included. If you choose to keep the abbreviation, you must get the associated number and in the first line also indicate the abbreviation. In Table 3, if the p-values are presented, it is redundant and the column with asterisks should be deleted.

Reviewer #2: A good piece of work demonstrating the impact of psychological stressors on the front line nurses during the pandemic.

It would be great if the authors and other researchers do further studies with other cultures and different healthcare settings.

6. PLOS authors have the option to publish the peer review history of their article (what does this mean?). If published, this will include your full peer review and any attached files.

Reviewer #1: No

Reviewer #2: **Yes: **Dr Lily Abedipour MD

---

## [Author Response · Author response to Decision Letter 0]

13 May 2022

Additional Editor Comments:

Thank you for your instructive comments and suggestions. We have revised the manuscript in adherence to your advice.

Abstract:

• Authors mentioned that “Our results showed an association between the HADS score and fear of COVID-19 and personality traits”- please consider mentioning whether this association is positive or negative.

Response: Thank you for your comment. Following Reviewer 1’s recommendation, we have conducted a logistic regression analysis instead of a liner regression analysis. Therefore, some of the results presented have also changed in the revised manuscript. We have attempted to describe the results concretely and clearly. In the abstract, we have reported the odd rations with 95% confident intervals as well as the values of R squared.

• Also, authors may consider mentioning the beta estimates with CI of multiple regression model.

Response: Thank you for your comment. We have conducted a logistic regression analysis instead of a multiple regression analysis. Therefore, we have mentioned the odd rations and its CI in the revised abstract.

Introduction:

• Citations are missing in several statements in the introduction. Please provide citations on all the facts.

Response: Thank you for your comment. We have added the references for all such statements .

• Fig 1 is blur, therefore losses readability.

Response: Thank you for your comment. We have reworked on Fig. 1. Although it still appears to be blurred in the PDF, you can download the original version of Fig 1. from the link. 

https://www.dropbox.com/s/y1zgg1g7w26ry83/Figure1.tiff?dl=0

Kindly provide suggestions regarding how we can present high-quality figures in the PDF as well.

Methods:

• What is the study design?

Response: Thank you for your comment. We have mentioned that this is a cross-sectional study in the first part of the “Methods” section.

• Please provide inclusion and exclusion criteria.

Response: Thank you for your comment. We have added the inclusion and exclusion criteria in the “Participants and procedures” section.

• This study was approved by the Research Ethics Committee of Z University (approval no. zzzz). Does this University exist? The ref. number does not look real.

Response: Thank you for your comment. We apologize for not presenting the name of the Research Ethics Committee correctly. We misunderstood the instructions regarding anonymized peer review and did not provide the name of the ethics committee in the manuscript. We have now presented the appropriate name and approval number of the committee.

• Please provide the Cronbach alpha of all measures used in this study.

Response: Thank you for your comment. As per your suggestion, we have added the Cronbach’s alpha of all the measures used in our study.

Results:

• Please split the table 3 into two separate table since it has two different outcomes.

Response: Thank you for your comment. As per your suggestion, we have modified the table to present the different outcomes. Additionally, we have conducted a logistic regression analysis instead of a multiple regression analysis, following Reviewer 1’s advice; the results of the logistic regression analysis are provided in Table 3.

• Authors may not require repeating the results what was shown in the tables. I’d suggest just writing the key results presented in the table.

Response: Thank you for your comment. As per your recommendation, we have modified the tables to present only the key results.

• I’m confused to understand the variables when I look at the table 1 and table 2. I see that authors presented categorical anxiety and depression scale in table 1 and continuous value in table 2. Please clarify.

Response: Thank you for your comment. We apologize for the lack of explanation on HADS. The HADS has a cut-off point for each subscale, depression, and anxiety. In Table 1, ‘Depression’ and ‘Anxiety’ presents each of the calculated HADS scores, while ‘Score of depression/anxiety’ indicates the number of people with depression/anxiety symptoms and people without symptoms. In Table 2, ‘Depression’ and ‘Anxiety’ represents the score of HADS. To avoid confusion, we have described the cut-off point of HADS.

Discussion:

• Authors need to elaborately discuss the study findings including clearly stating the interpretation of the results and what was found in the earlier studies.

• Furthermore, discussion is warranted based on theoretical aspect.

Response: Thank you for your instructive comments. We have revised the discussion section entirely to explain the findings of our study. In particular, we have focused on the relationship between FCV-19S, the Big Five personality traits, and depression/anxiety in our discussion. We have also compared the findings with those of previous studies.

• I’d not use ‘mental health’ in the discussion rather use anxiety and depression.

Response: Thank you for your comment. We agree with your suggestion. To express clearly, we have modified the term, “mental health” to “anxiety and depression.”

• There are quite a number of limitations that authors should mention. For instance, generalizability, limitations of using self-reported data, causality...potential biases etc.

Response: Thank you for your comment. We have added the limitations of the study such as those related to the measures used, self-reported questionnaires, procedure to distribute them, and sampling bias.

Conclusion:

• Authors mentioned that “The results of this study indicated that the fear of COVID-19 and the Big Five personality traits could predict the levels of depression and anxiety using the HADS.” – I’m not sure what this means?

o I understand the fear of covid-19 increases anxiety and depression among the Nurses? Is this correct? If yes, please provide a clear recommendation rather than using ambiguous terms.

o I’m lost about the conclusion on the Big Five personality traits. I am not sure what authors are trying to say? Is it that those 5 personality traits predict increased anxiety and depression, sorry if I got it wrong? Please help me understand the conclusion.

Response: Thank you for your comment. We apologize that our explanations in the “Conclusion” section were not clear. The findings of our study presented neuroticism to be a common risk factor of depression and anxiety in this revised manuscript. Additionally, the FCV-19S score was considered a risk factor only for anxiety. We have modified the “Conclusion” section to describe these results clearly.

Reviewers' comments:

Reviewer #1

Thank you for your careful review, instructive comments, and suggestions. We have revised the manuscript to incorporate your suggestions.

- The abstract contains excessive information on the method (questionnaires distributed, answers received, ...), only the final number of participants should be indicated since the rest of the information is already exposed in the manuscript. Instead, the implications and/or interest of the results obtained should be added.

Response: Thank you for your comment. Following your advice, we have modified the abstract to present less information on the methods used and added more information on the results of the logistic regression analysis.

- The use of the keyword “mental health”, even in the title, is too broad. With it, it is expected to find the evaluation of symptomatology corresponding to more disorders, but the manuscript is limited to exploring depression and anxiety. It is recommended to specify the symptomatology studied.

Response: Thank you for your comment. We agree with your suggestion. To express clearly, we have modified “mental health” to “anxiety and depression.” Furthermore, we have modified the title to “The role of Big Five personality traits and the fear of COVID-19 predict depression and anxiety among Japanese nurses caring for COVID-19 patients: A cross-sectional study in Wakayama prefecture.”

- The authors comment that in Wakayama prefecture the incidence of Covid-19 is low and detail the third wave situation at the time of collecting information (page 9, lines 110-121), this information can be reduced without losing the content essential.

Response: Thank you for your comment. As per your suggestion, we have reduced the number of descriptions explaining the situation at the time this study was conducted.

- The use of Big Five personality, although widespread in the international community, is based on lexical aspects. There are other questionnaires with a robust psychobiological basis that are more suitable in the field of mental health work. Among them, Zuckerman's ZKPQ and Cloninger's TCI-R stand out. In the limitations section, this limitation should be added with the inclusion of the paper by Muro et al. (Chronobiology International, 28 (7), 690-696) in normal population and from Rio-Martinez et al. (Journal of Clinical Medicine, 9(6): 1876) in patients with mental disorders, the reading of both works in relation to the traits of personality of both questionnaires will provide the authors with this vision. The interest of considering more aspects of temperament or character, in the case of the TCI-R, is of the utmost importance. Introducing this approach also makes it possible to criticize the fact that the use of the Big Five personality presents relationships that are nonspecific in relation to the symptomatology studied.

Response: Thank you for your suggestions. These articles were extremely helpful for us. As a result, we realized that there exists a limitation in the Big Five models. We understood your suggestion that the findings we have presented could be theoretically explained from the two domains of Temperament and Character and our results might be supported by the psychobiological basis if we used these. We have added this in the “Study limitation” section.

- Measures. In this section, mention should not only be made of the reliability published in previous works, but also the detail of the reliability in the sample studied (FCV-19S, BFS and HADS) is required.

Response: Thank you for your comment. Following your suggestion, we have added the information on the reliability and details of all scales, as discussed in previous studies.

- In reference to the cut-off points of the HADS, the interest is focused on differentiating between “non-anxiety or depression” (0-7) and “possible case” (8 or more). Actually, the authors do not then work in their analyzes with the three possible groups indicated in the measures section (page 11). In table 1 it is not correct to define the depression and anxiety scores as low / high, I suggest adopting presence/absence of symptoms or yes/no possible case.

Response: Thank you for your comment. A goal of our study is to screen and provide early support to nurses suffering from depression or anxiety. In this study, we attempt to find the possibilities of the occurrences of depression or anxiety. Therefore, it might not be appropriate to explain the cut-off points scoring with regard to the three groups in light of our purpose. Hence, we have modified the explanation regarding HADS; we used the cut-off points to divide the two groups of “non-anxiety or depression” and “possible case.” Additionally, we have modified the expression of the possible cases of depression or anxiety in Table 1, following your advice.

- The choice of linear regression analysis is not the most appropriate. To establish the existence or not of an association between personality traits and symptoms of anxiety and depression (presence/absence or yes/no possible case), the approach requires logistic regressions. Since concluding that, for example, the more neuroticism the score is higher, it does not mean that a high neuroticism is really associated with scores for the presence of symptoms.

Response: Thank you for your suggestion. Reconsidering the goal of our study, we realized that logistic regression analysis would be the more appropriate method for this study, as you had suggested. Consequently, we reanalyzed the data using this method and modified their related descriptions in the “Methods,” “Results,” “Discussion,” and “Conclusion” sections. According to the findings of this analysis, neuroticism was the only common risk factor of depression and anxiety while the FCV-19S score, age, and sex were risk factors only for anxiety. 

- In the results text (pages 14-15) the numerical data already provided in Table 1 are repeated. The two paragraphs should be rewritten in a more qualitative way and only highlighting some data of more interest.

Response: Thank you for your comment. Following your suggestions, we have modified the results to present the critical data.

- Presentation of data in the tables can be improved. In Table 2 the first line can be deleted and the results could be better interpreted if the full name of the variables is included. If you choose to keep the abbreviation, you must get the associated number and in the first line also indicate the abbreviation. In Table 3, if the p-values are presented, it is redundant and the column with asterisks should be deleted.

Response: Thank you for your comment. As per your suggestion, we have revised Table 2; the first line was modified, and the full names of the variables were included.

Reviewer #2: A good piece of work demonstrating the impact of psychological stressors on the front line nurses during the pandemic. It would be great if the authors and other researchers do further studies with other cultures and different healthcare settings.

Response: Thank you for your comments and suggestions. Following the vaccination protocols, the COVID-19 situation in Japan is changing, and we will keep investigating this aspect further.

---

## [Decision Letter · Decision Letter 1]

20 Jun 2022

PONE-D-22-04225R1The Big Five personality traits and the fear of COVID-19 predict depression and anxiety among Japanese nurses caring for COVID-19 patients: A cross-sectional study in Wakayama prefecturePLOS ONE

Dear Dr. Odachi,

Thank you for submitting your manuscript to PLOS ONE. After careful consideration, we feel that it has merit but does not fully meet PLOS ONE’s publication criteria as it currently stands. Therefore, we invite you to submit a revised version of the manuscript that addresses the points raised during the review process.

We look forward to receiving your revised manuscript.

Kind regards,

Fakir Md Yunus, PhD, MSC, MPH, MBBS

Academic Editor

PLOS ONE

Additional Editor Comments:

Many thanks for your patience in responding to the authors' comments. I've some last-minute concerns about the analysis.

(1) I am not sure I clearly understand the reason for using logistic regression. For instance, authors mentioned at the end of intro that "The purpose of this study was to clarify the relationship between the mental health problems of depression and anxiety in nurses who had to deal with patients afflicted with COVID-19 or was suspected COVID-19, the fear of COVID-19, and the Big Five personality traits." But based on the analysis, data nature and conclusion, it reads to me that authors investigated the risk/likelihood of depression and anxiety predicted by 5 traits, age and sex. Since the measures were collected as scale, I am not sure why authors considered to recode the cont. variable to cat. variable. I may have missed the point here.

(2) May I also kindly request authors to clarify why not 'linear mixed model with hospitals as random effect" is the best analysis approach for this study.

(3) Regards to sex variable, I suggest authors to present the male and female in the table rather presenting as foot note the cat.

(4) why sex is missing in the table 4.

(5) Please consider providing some examples of each of the scales used in this study and state the response option. I see that authors mentioned that they have used 0-3 scale and what does the 0 represents.....

(6) Please consider providing the scoring calculation of the score used in this study. For instance, authors mentioned that HADS has 7-items for depression. Is it the mean of 7 item or the total score treated as depression.

(7) Please kind provide ref. of articles that used categorical HADS scale. Where does the cut-off for " is a score of more than 8" comes from.

(8) Write full form of FCV19S in the tables.

(9) P8 L 119; what does "......32% are aged" mean.

(10) English editing is necessary.

Reviewers' comments:

Reviewer's Responses to Questions

**Comments to the Author**

1. If the authors have adequately addressed your comments raised in a previous round of review and you feel that this manuscript is now acceptable for publication, you may indicate that here to bypass the “Comments to the Author” section, enter your conflict of interest statement in the “Confidential to Editor” section, and submit your "Accept" recommendation.

Reviewer #1: (No Response)

Reviewer #2: All comments have been addressed

2. Is the manuscript technically sound, and do the data support the conclusions?

Reviewer #1: Yes

Reviewer #2: Yes

3. Has the statistical analysis been performed appropriately and rigorously? 

Reviewer #1: Yes

Reviewer #2: I Don't Know

4. Have the authors made all data underlying the findings in their manuscript fully available?

Reviewer #1: Yes

Reviewer #2: Yes

5. Is the manuscript presented in an intelligible fashion and written in standard English?

Reviewer #1: Yes

Reviewer #2: Yes

6. Review Comments to the Author

Reviewer #1: The presentation of data in tables can be improved. In Table 1 it is enough to detail the "yes". In Table 2 the line for the correlation between age and age should be deleted. In general, authors should carefully review the details of the tables.

Reviewer #2: It seems the authors have tried to address the comments and enquiries of the reviewer accordingly and have made changes even to their methods. I think this work deserves being published.

7. PLOS authors have the option to publish the peer review history of their article (what does this mean?). If published, this will include your full peer review and any attached files.

Reviewer #1: No

Reviewer #2: No

---

## [Author Response · Author response to Decision Letter 1]

11 Jul 2022

We would like to express our sincere gratitude to the editor and reviewers. We have revised our manuscript to incorporate your valuable suggestions.

Additional Editor Comments:

Thank you for your instructive comments and suggestions. We have now revised the manuscript as per your advice.

(1) I am not sure I clearly understand the reason for using logistic regression. For instance, authors mentioned at the end of intro that "The purpose of this study was to clarify the relationship between the mental health problems of depression and anxiety in nurses who had to deal with patients afflicted with COVID-19 or was suspected COVID-19, the fear of COVID-19, and the Big Five personality traits." But based on the analysis, data nature and conclusion, it reads to me that authors investigated the risk/likelihood of depression and anxiety predicted by 5 traits, age and sex. Since the measures were collected as scale, I am not sure why authors considered to recode the cont. variable to cat. variable. I may have missed the point here.

Response: Thank you for your comment. When we changed the method of the statistical analysis, we also changed the study’s purpose to investigate the risk of depression and anxiety in nurses, predicted by the fear of COVID-19 and the Big Five personality traits as described by you. It was because HADS was developed for screening depression and anxiety. Therefore, we thought it would be appropriate to analyze it as a categorical variable. This could show more clearly to what extent the scores of the Big Five scale and FCV-19S predict the risk for depression and anxiety. We noticed them due to the reviewer’s suggestions. We apologize for not changing the new purpose of the study. We have now modified the purpose appropriately to match the logistic regression analysis. 

(2) May I also kindly request authors to clarify why not 'linear mixed model with hospitals as random effect" is the best analysis approach for this study.

Response: Thank you for your comment. As described above, we have changed the statistical method because we have considered that it would more explicitly present to readers the extent to which the independent variables affect nurses’ mental health problems. In the result of multiple linear regression, we thought it was a little difficult to understand how much of a problem the fluctuations in HADS scores brought by independent variables are regarded as clinical symptoms. We have added the reason to use logistic regression analysis in the “Statistical Analyses” section.

(3) Regards to sex variable, I suggest authors to present the male and female in the table rather presenting as foot note the cat.

Response 3: Thank you for your comment. As per your recommendation, we have now moved the label from the footnotes to the variable cells in Tables 3 and 4. 

(4) why sex is missing in the table 4.

Response 4: Thank you for your comment. This was a simple mistake, and it seems that we have included it in another table that analyzed with this variable removed. We apologize for including the incorrect table. Thank you for pointing this out. Additionally, the values of the result of the logistic analysis in this manuscript text (lines 216-225) have been not modified because we had already referred to Table 4 in the previous manuscript.

(5) Please consider providing some examples of each of the scales used in this study and state the response option. I see that authors mentioned that they have used 0-3 scale and what does the 0 represents.....

Response 5: Thank you for your comment. We have added the explanation that high HADS scores mean more frequent symptoms or stronger symptoms. We have also included some examples of the question and the scoring method for its responses. 

(6) Please consider providing the scoring calculation of the score used in this study. For instance, authors mentioned that HADS has 7-items for depression. Is it the mean of 7 item or the total score treated as depression.

Response 6: Thank you for your comment. The total score was derived adding up the scores of all the scales used in our study (HADS, FBS, FCV-19S). The total scores of each scale were used in the logistic regression analysis. We have added this in the explanations of each scale.

(7) Please kind provide ref. of articles that used categorical HADS scale. Where does the cut-off for " is a score of more than 8" comes from.

Response 7: Thank you for your comment. We apologize for the lack of reference. We have added the reference to explain the cut-off point.

(8) Write full form of FCV19S in the tables.

Response 8: Thank you for your comment. We have added the full form of FCV-19S in the footnotes of all the tables.

(9) P8 L 119; what does “…...32% are aged" mean.

Response 9: Thank you for your comment. In the sentence, it was meant that the aging rate of the population of Wakayama prefecture was 32%. We have now modified the explanation to be appropriate. We described this value as an indicator of the regional character of Wakayama prefecture that there is a relatively large older population because older age is considered a risk factor associated with disease severity.

(10) English editing is necessary.

Response 10: Thank you for your comment. We have opted for a professional English editing service to further enhance the quality of our entire manuscript.

Reviewers' comments:

Reviewer #1

Thank you for your careful review, instructive comments, and suggestions. We have revised the manuscript to incorporate your suggestions.

- The presentation of data in tables can be improved. In Table 1 it is enough to detail the "yes". In Table 2 the line for the correlation between age and age should be deleted. In general, authors should carefully review the details of the tables.

Response: Thank you for your suggestion. We have deleted this line from Table 2 as per your suggestion. 

Reviewer #2

- It seems the authors have tried to address the comments and enquiries of the reviewer accordingly and have made changes even to their methods. I think this work deserves being published.

Response: Thank you for your supportive comments.

---

## [Editor Report · Decision Letter 2]

22 Aug 2022

PONE-D-22-04225R2The Big Five personality traits and the fear of COVID-19 in predicting depression and anxiety among Japanese nurses caring for COVID-19 patients: A cross-sectional study in Wakayama prefecturePLOS ONE

Dear Dr. Odachi,

Thank you for submitting your manuscript to PLOS ONE. After careful consideration, we feel that it has merit but does not fully meet PLOS ONE’s publication criteria as it currently stands. Therefore, we invite you to submit a revised version of the manuscript that addresses the points raised during the review process.

We look forward to receiving your revised manuscript.

Kind regards,

Fakir Md Yunus, PhD, MSC, MPH, MBBS

Academic Editor

PLOS ONE

Additional Editor Comments:

Dear Authors,

Many thanks for your kind responses. The paper is now much improved in my opinion; however there are certain things still needed to be clear. I am sorry that I had to get back to you again.

• The clarification for using multivariate logistic regression instead of mixed-effects logistic regression is not convincing. Please clarify the reason for not controlling random effect in the model A and Model B. Doesn’t hospital can be considered as a “random effect” variable which will account the variability of nurses’ experience across 13 hospitals? For example, isn’t it possible that nurses from a certain study hospital might experience more stress than others when exposed to the same level of stress.

• Table 1: is it mean+/- SD or mean+/- SE. I think authors are using SD; however, I see in the table 3 and 4 authors used SE. Isn’t author should be consistent with SE across the paper.

• P 14, L 216 authors wrote “Table 3 and Table 4 present the results of the multiple regression analysis for model A and B, respectively” isn’t should be multivariate logistic regression. I understand multiple regression represents linear regression.

• I think it might be useful to provide the VIF of the variables at the end of the table 2.

• It would be useful to merge table 3 and table 4 so that it can be easily understood by the reader.

• First few lines of the discussion does not match with the revised study objective.

• Authors rightly cited previous studies in support to the study findings; however, I noticed that ‘Why” is missing throughout the discussion. Plus, how the study findings differed from the theory is essential to discuss.

Thank you again.
---

## [Author Response · Author response to Decision Letter 2]

19 Sep 2022

Dear Dr. Yunus,

Thank you for giving us the opportunity to submit a revised draft of our manuscript titled “The Big Five personality traits and the fear of COVID-19 in predicting depression and anxiety among Japanese nurses caring for COVID-19 patients: A cross-sectional study in Wakayama prefecture” to PLOS ONE. We appreciate the time and effort that you and the reviewers have dedicated to providing your valuable feedback on my manuscript. We are grateful to the reviewers for their insightful comments on my paper. We have been able to incorporate changes to reflect most of the suggestions provided by the reviewers. We have highlighted the changes within the manuscript. 

Here is a point-by-point response to the reviewers’ comments and concerns.

Editor Comments:

(1) The clarification for using multivariate logistic regression instead of mixed-effects logistic regression is not convincing. Please clarify the reason for not controlling random effect in the model A and Model B. Doesn’t hospital can be considered as a “random effect” variable which will account the variability of nurses’ experience across 13 hospitals? For example, isn’t it possible that nurses from a certain study hospital might experience more stress than others when exposed to the same level of stress.

Response: Following your suggestion, we have added the explanation for performing multivariate logistic regression analysis below this sentence. Additionally, we have compared the scores of key scales (the HADS, the FCV-19S, and the BFS) between hospitals and have found no statistically significant differences. Therefore, it is considered that the variable “hospital” has a limited impact on the outcomes of this study. However, since the number of participants was unbalanced among the hospitals, it is possible that there might be differences in nurses’ experiences. We have added this in the limitations section.

Page 21, Lines 368–370 

“Additionally, the number of participants across hospitals was unbalanced, and it is possible that the differences in nurses’ experiences between hospitals were not reflected in these results.”

(2) Table 1: is it mean+/- SD or mean+/- SE. I think authors are using SD; however, I see in the table 3 and 4 authors used SE. Isn’t author should be consistent with SE across the paper.

Response: Following your suggestion, we have modified our paper to use SE consistently. We have modified table 1 and parts of the method and results related to this.

(3) P 14, L 216 authors wrote “Table 3 and Table 4 present the results of the multiple regression analysis for model A and B, respectively” isn’t should be multivariate logistic regression. I understand multiple regression represents linear regression.

Response: We apologize for the error in the previous version of the manuscript. Following your feedback, we have modified it to “multivariate logistic regression.”

Page 14, Lines 224–225

“Table 3 presents the results of the multivariate logistic regression analysis for models A and B”

(4) I think it might be useful to provide the VIF of the variables at the end of the table 2.

Response: Although we agree with your suggestion, providing the VIF of the variables in Table 2 would make it too large. Therefore, we have added the VIFs in table 3, which presents the results of the logistic multivariate analysis.

(5) It would be useful to merge table 3 and table 4 so that it can be easily understood by the reader.

Response: Following your suggestion, we have merged Tables 3 and 4 into a new table, that is, Table 3.

(6) First few lines of the discussion does not match with the revised study objective.

Response: We have modified this sentence to match the study objective. 

Page 16, Lines 240–241

“This study aimed to assess the risk of depression and anxiety among nurses who have experience of caring for patients afflicted with COVID-19 in Japan.”

(7) Authors rightly cited previous studies in support to the study findings; however, I noticed that ‘Why” is missing throughout the discussion. Plus, how the study findings differed from the theory is essential to discuss. 

Response: Following your feedback, we noticed that we did not thoroughly explain the result that extraversion has no impact on depression and anxiety. Therefore, we have provided an explanation for this result in the discussion section.

Page 19, Lines 309–312

“The pandemic has caused movement restrictions and people to avoid contact with each other. This may have brought about inadequate coping strategies for people with high extraversion. Therefore, it is considered that extraversion has not positively influenced nurses’ risk for depression and anxiety.”

---

## [Editor Report · Decision Letter 3]

14 Oct 2022

The Big Five personality traits and the fear of COVID-19 in predicting depression and anxiety among Japanese nurses caring for COVID-19 patients: A cross-sectional study in Wakayama prefecture

PONE-D-22-04225R3

Dear Dr. Odachi,

We’re pleased to inform you that your manuscript has been judged scientifically suitable for publication and will be formally accepted for publication once it meets all outstanding technical requirements.

Kind regards,

Fakir Md Yunus, PhD, MSC, MPH, MBBS

Academic Editor

PLOS ONE

Additional Editor Comments (optional):

Many thanks. It'd be great if the manuscript go through English editing.

Reviewers' comments:

<quillbot-extension-portal></quillbot-extension-portal>

---

## [Editor Report · Acceptance letter]

20 Oct 2022

PONE-D-22-04225R3 

The Big Five personality traits and the fear of COVID-19 in predicting depression and anxiety among Japanese nurses caring for COVID-19 patients: A cross-sectional study in Wakayama prefecture 

Dear Dr. Odachi:

I'm pleased to inform you that your manuscript has been deemed suitable for publication in PLOS ONE. Congratulations! Your manuscript is now with our production department. 

Kind regards, 

on behalf of

Dr. Fakir Md Yunus 

Academic Editor

PLOS ONE